**Data Availability Statement:** All relevant data are within the manuscript and its Supporting Information files.

**Funding:** This work was supported by grants from the Research Council of Norway (project 275053 to

# Injection of prototypic celiac anti-transglutaminase 2 antibodies in mice does not cause enteropathy

**Christian B. Lindstad**[1,2], **M. Fleur du Pré**[1,3], **Jorunn Stamnaes**[1,2,3], **Ludvig M. Sollid**[1,2,3]*

**1** K.G. Jebsen Coeliac Disease Research Centre, University of Oslo, Oslo, Norway, **2** Department of Immunology, University of Oslo, Oslo, Norway, **3** Department of Immunology, Oslo University Hospital, Oslo, Norway

* l.m.sollid@medisin.uio.no

## Abstract

### Background

Celiac disease is an autoimmune enteropathy driven by dietary intake of gluten proteins. Typical histopathologic features are villous flattening, crypt hyperplasia and infiltration of inflammatory cells in the intestinal epithelium and lamina propria. The disease is hallmarked by the gluten-dependent production of autoantibodies targeting the enzyme transglutaminase 2 (TG2). While these antibodies are specific and sensitive diagnostic markers of the disease, a role in the development of the enteropathy has never been established.

### Methods

We addressed this question by injecting murine antibodies harboring the variable domains of a prototypic celiac anti-TG2 immunoglobulin into TG2-sufficient and TG2-deficient mice evaluating for celiac enteropathy.

### Results

We found no histopathologic abnormalities nor clinical signs of disease related to the injection of anti-TG2 IgG or IgA.

### Conclusions

Our findings do not support a direct role for secreted anti-TG2 antibodies in the development of the celiac enteropathy.

## Introduction

Celiac disease is an autoimmune enteropathy driven by ingestion of gluten proteins. The celiac lesion is characterized by villous flattening, crypt hyperplasia and infiltration of inflammatory cells. Extraintestinal manifestations include skin disease, anemia, osteopenia, neurological

L.M.S.), the European Commission (project ERC-2010-Ad-268541 to L.M.S.), and the University of Oslo. The funders had no role in study design, data collection and analysis, decision to publish, or preparation of the manuscript.

**Competing interests:** The authors have declared that no competing interests exist.

symptoms and obstetric complications. The enzyme transglutaminase 2 (TG2) has a crucial role in the disease, both by catalyzing deamidation of gluten peptides into immunogenic T cell epitopes [1], and by being the target of a disease-specific autoantibody response [2]. Anti-TG2 IgA and IgG are sensitive and specific markers of the disease [3,4]. These antibodies disappear rapidly from the blood upon initiation of a gluten free diet [5]. When absent in serum, the antibodies may still be found in the intestinal tissue [6,7].

Anti-TG2 antibodies have previously been suspected to contribute to enteropathy as well as extraintestinal manifestations of celiac disease [8]. In support of this, the antibodies have been reported to deposit extracellularly both in the intestine and at extraintestinal sites [9,10]. Serum titers correlate with the degree of enteropathy [11]. Yet, a group of individuals defined as "potential celiac disease" have anti-TG2 in serum and in the intestinal mucosa despite normal mucosal histology [12].

If anti-TG2 antibodies play a role in development of enteropathy, they would be an interesting therapeutic target. *In vitro* studies have reported numerous biological effects of anti-TG2 antibodies, including inhibiting differentiation, inducing proliferation or reducing attachment of intestinal epithelial cells, as reviewed [8]. The *in vivo* evidence supporting a role of secreted anti-TG2 antibodies in celiac enteropathy is, however, scarce and inconclusive. When generating an anti-TG2 response *in vivo* by immunizing with TG2 [13] or expression of mini-antibodies using virus vectors [14], no significant pathology was observed. Kalliokoski et al. injected celiac IgA-deficient serum, total IgG or recombinant monoclonal anti-TG2 mini-antibodies [15,16]. They observed minor histologic changes and in one study minor clinical effects. Limitations of these studies are the use of non-physiological antibodies (mini-bodies or polyclonal human serum antibodies) as well as immunocompromized mouse strains. To conclusively address whether anti-TG2 antibodies as found in the serum of celiac patients play a direct role in the development of enteropathy, we injected TG2-sufficient and TG2-deficient mice with murine IgG or IgA harboring the variable domains of the prototypic celiac anti-TG2 antibody 679-14-E06 (from here denoted 14E06) [17]. We observed no evidence of enteropathy nor clinical signs of disease. Thus, this study does not support a direct role for anti-TG2 antibodies in development of celiac enteropathy.

## Materials and methods

### Generation of murine 14E06 antibodies

Hybridomas producing monoclonal murine antibodies with 14E06 variable domains were generated from naive B cells of 14E06 immunoglobulin knock-in mice as described [18]. The 14E06 antibody has equal affinity (5 nM) to mouse and human TG2 [18]. Antibodies were purified from culture supernatants using HiTrap protein L columns (GE), buffer exchanged to PBS and sterile filtered before storage at -20°C until use.

### Mice

C57Bl/6 mice were purchased from Janvier Labs. *Tgm2*$^{-/-}$ mice on C57Bl/6 background [19] were kindly provided by G. Melino and bred in-house. Mice were age and sex-matched between groups and included in the experiment at 6 or 8 weeks of age. Each experimental group was split evenly between cages. Injections were performed cage-by-cage. Mice were kept at the Department of Comparative Medicine, Oslo University Hospital, Rikshospitalet (Oslo, Norway) under specific pathogen-free conditions. They were inspected daily by attending staff during the experiments and weighed at least every other day. All animal experiments were pre-approved by the Norwegian Food Safety Authority (Mattilsynet).

## Experimental procedures and collection of samples

IgA or a mix of IgG2b and IgG2c were diluted in sterile PBS. For each injection, 200 μL was injected in the tail vein. Blood samples were collected from the lateral saphenous vein on day 0, 10 and 20 (Fig 1). For four IgA-injected mice, the third sample was taken on day 16 or 17, and a fourth sample on day 20 was obtained by postmortem cardiac puncture. Blood was allowed to coagulate for 1–2 hours, centrifuged at 900 g for 14 min at 4˚C and serum was stored at -20˚C. At the end of the experiment, the small intestine was extracted and the proximal 2 cm discarded. Boluses of feces were gently flushed out with ice-cold PBS. Samples from corresponding gut segments were fixed in 10% neutral buffered formalin (Sigma) for 24 hours, dehydrated and embedded in paraffin. The automated Tissue-Tek Paraform Sectionable Cassette System (Sakura) was used with orientation gels to ensure proper orientation.

## Histology and immunohistochemistry

Paraffin embedded samples were cut into 2.5 μm sections. In hematoxylin/eosin-stained sections, villus height (Vh), crypt depth (Cd) and villus height/crypt depth ratio (Vh/Cd ratio) were measured only for well oriented villus-crypt pairs. Spanning at least three gut pieces, the mean values of the five Vh/Cd pairs with the longest villi were reported. If five valid measurements could not be obtained from one gut segment, the segment was excluded from the analysis. Number of excluded data points for duodenum/ileum in each group: WT IgG: 4/1, $Tgm2^{-/-}$ IgG: 6/1, WT IgA: 3/1, WT PBS: 2/0, $Tgm2^{-/-}$ PBS 1/0. For intraepithelial lymphocyte (IEL) counts, sections were stained for CD3 and counterstained with hematoxylin. As primary antibody, rabbit monoclonal anti-CD3, (SP7, Abcam) was used at 1:100. Samples were pretreated with Dako Target Retrieval Solution Citrate pH 6 (Agilent Technologies). For detection, rabbit on Rodent HRP (Biocare Medical) was used followed by development with 3,3′-diaminobenzidine. CD3+ IELs were expressed per 100 epithelial cells in a hotspot villus, reporting the mean of three measurements from thee different gut pieces. Slides were scanned with Pannoramic

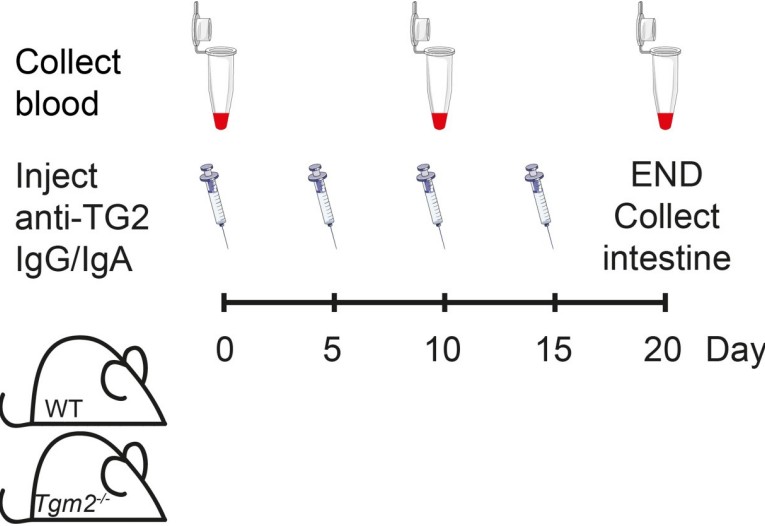

**Fig 1. Overview of experimental setup.** Anti-TG2 was administered by intravenous injection at the indicated time points. Either a mix of IgG2b and IgG2c (100 μg each) or IgA (400 μg) or PBS was given each time. Blood was collected at indicated time points prior to injections. At the end of the experiment, samples of the small intestine were fixed in formalin or embedded in OCT and frozen. The figure was created using elements from Servier medical arts (www. smarts.servier.com).

Midi and analyzed with Case Viewer (both 3DHISTECH) blinded to the investigator. Evaluation criteria were defined *a priori*.

## Immunofluorescence

Unfixed small intestine was embedded in optimal cutting temperature (OCT) and snap frozen in liquid nitrogen. Six μm sections were adhered to SuperFrost slides by thaw-mounting and air-dried. To demonstrate binding of hybridoma-derived mouse 14E06 antibodies to mouse and human TG2, 6 μm unfixed tissue sections from WT mouse small intestine, or $Tgm2^{-/-}$ mouse small intestine pre-incubated with recombinant human TG2 or recombinant mouse TG2 (7 μg/ml), were stained with 3 μg/ml 14E06 mouse IgG2c followed by detection with donkey anti mouse IgG-Cy3 (Jackson ImmunoResearch) (S1 Fig). To assess co-localization between injected IgG and endogenous TG2, unfixed small intestinal tissue sections were blocked in 1.25% IgG-free BSA (Jackson Immunoresearch) in PBS and stained with goat-anti-mouse-IgG (Jackson ImmunoResearch) and rabbit-anti-mouse-TG2 (custom made antibody from Pacific Immunology) followed by detection with donkey anti-goat Alexa Fluor 488 (Jackson ImmunoResearch) and donkey anti-rabbit Cy3 (Jackson ImmunoResearch). To quantify and assess tissue deposition of injected IgG, unfixed sections were stained with anti-mouse-IgG2b-biotin and anti-mouse-IgG2c-biotin (both SouthernBiotech) (both at 3 μg/ml, as a mix or separately) followed by Streptavidin-Cy3 (2.5 μg/mL) (GE Lifesciences). Slides were counterstained with 40,6-diamidino-2-phenylindole (DAPI) and mounted with ProLong Diamond Antifade Mountant (ThermoFisher). Slides were imaged on an inverted Nikon fluorescence microscope (Nikon Eclipse Ti-S; Nikon, Tokyo, Japan) and images were processed in Fiji (ImageJ) [20]. Subepithelial antibody deposits were quantified in the small intestine of IgG-injected WT (n = 4) and $Tgm2^{-/-}$ mice (n = 4) as well as PBS injected WT (n = 1) and $Tgm2^{-/-}$ mice (n = 1). Fluoresence intensity was quantified from 4–8 villi per image and 1–2 images were anlyzed per mouse. Fluorescence intensity was measured in FIJI from unprocessed images aquired with identical microscope settings. Subepithelial regions of interest were defined using the freehand tool (linewith 5 pixels for 20x images and 10 pixels for 10x images) and integrated density was measured. Integrated density from a region drawn within the epitelial cell layer of the same villus was subtracted as background.

## ELISA to evaluate anti-TG2 titers in serum

ELISA plates (Nunc) were coated with 5 μg/mL recombinant human TG2 [21] in PBS at 4˚C overnight. After washing and blocking, plates were incubated with dilutions of mouse serum (1.5 hours at room temperature) followed by biotinylated goat anti-mouse IgG2b, IgG2c or IgA (SouthernBiotech, 1.5 hours at room temperature), then alkaline phosphatase-conjugated streptavidin (SouthernBiotech, 0.5 hours at room temperature) before development with phosphatase-substrate (Sigma). Optical density was determined at 405 nm. Absolute concentrations were estimated by comparing with dilutions of antibody and interpolating from standard curves.

## Statistical methods and data visualization

Statistical comparisons and data visualization were done using GraphPad Prism 9.3.1 (Graph-Pad Software). For comparisons, individual Mann-Whitney tests were used. $P < 0.05$ was considered statistically significant, and no correction for multiple testing was applied. The study was powered to detect differences of >1 for Vh/Cd ratio and >10 for IEL counts with $\alpha = 0.05$ and $\beta = 0.20$.

## Results

### Generation of anti-TG2 antibodies and choice of isotypes

The patient-derived 14E06 is a prototypic celiac anti-TG2 antibody [17,22]. Murine antibodies harboring the 14E06 variable domains were generated using hybridoma technology [18]. To maximize the chances of revealing a potential inflammatory effect, we chose to inject the main experimental groups with a mix of 14E06 IgG2b and IgG2c (see discussion). Based on the fact that the clinical presentation of IgA deficient celiac patients is similar to that of IgA-sufficient patients [23–25], we regarded IgA-injected mice mainly as a control group.

### Overview of experimental setup and confirmation of injected anti-TG2 antibodies in serum and intestinal tissue

Experimental setup is outlined in Fig 1. Data were pooled from two independent experiments. Antibodies were injected intravenously at day 0, 5, 10 and 15. The main groups consisted of wild-type (WT, n = 14) and $Tgm2^{-/-}$ mice (n = 12) that received a mix of 100 µg IgG2b and 100 µg IgG2c each time. Additional groups included WT mice that received 400 µg IgA (n = 8) or PBS (n = 6), or $Tgm2^{-/-}$ mice that received PBS (n = 2). A higher dose of IgA was chosen because of short serum half-life [26]. In IgG-injected mice, high serum levels of both isotypes were detected on day 10 and 20 (Fig 2A and 2B). There were no statistically significant differences in serum levels between IgG-injected WT and $Tgm2^{-/-}$ groups. Surprisingly, no TG2-specific IgA was detected in serum on day 10 or 20 (Fig 2C). To confirm presence of injected IgA, blood was collected shortly after the 3rd antibody injection in a few mice. Small amounts of TG2-specific IgA could be detected in serum of 2/2 mice on day 16, while traces were detected in 1/2 mice on day 17 (Fig 2C), indicating rapid clearance. There was no reactivity to TG2 in serum of PBS-injected mice (Fig 2A–2C). Immunofluorescence staining of unfixed small

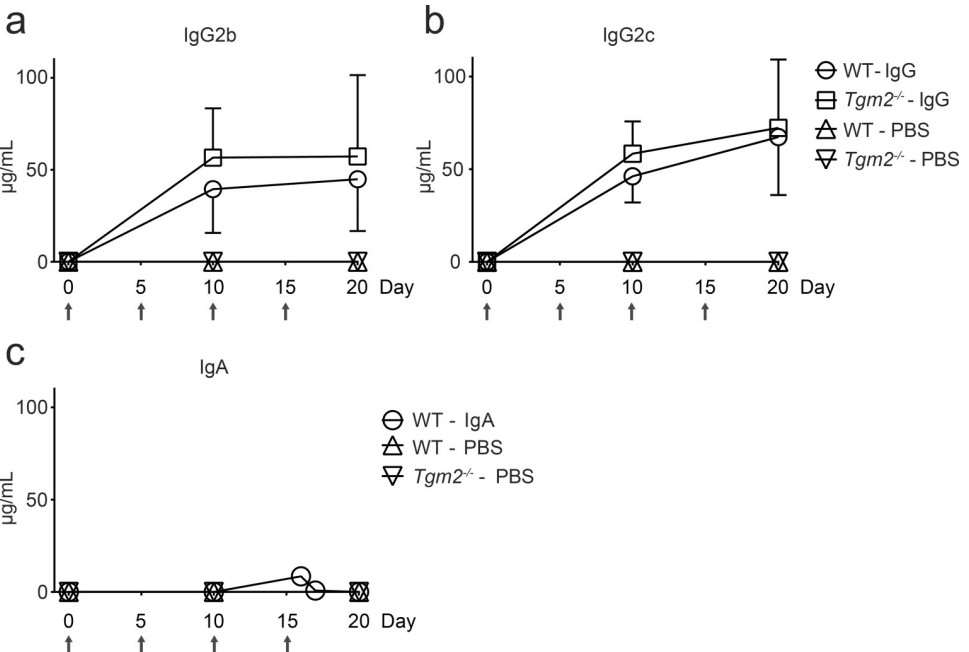

**Fig 2. Anti-TG2 in serum.** Serum was obtained on day 0, 10 and 20 and analyzed for anti-TG2 antibodies by ELISA. Absolute concentrations of IgG2b (a), IgG2c (b) and IgA (c) were estimated by interpolating from standard curves. For four IgA-injected mice, the third serum sample was taken either on day 16 (n = 2) or day 17 (n = 2) and a fourth sample was taken on day 20 by postmortem cardiac puncture. Arrows indicate time of antibody injections. Dots and bars represent mean +/- SD. Data represent all mice from the two independent experiments.

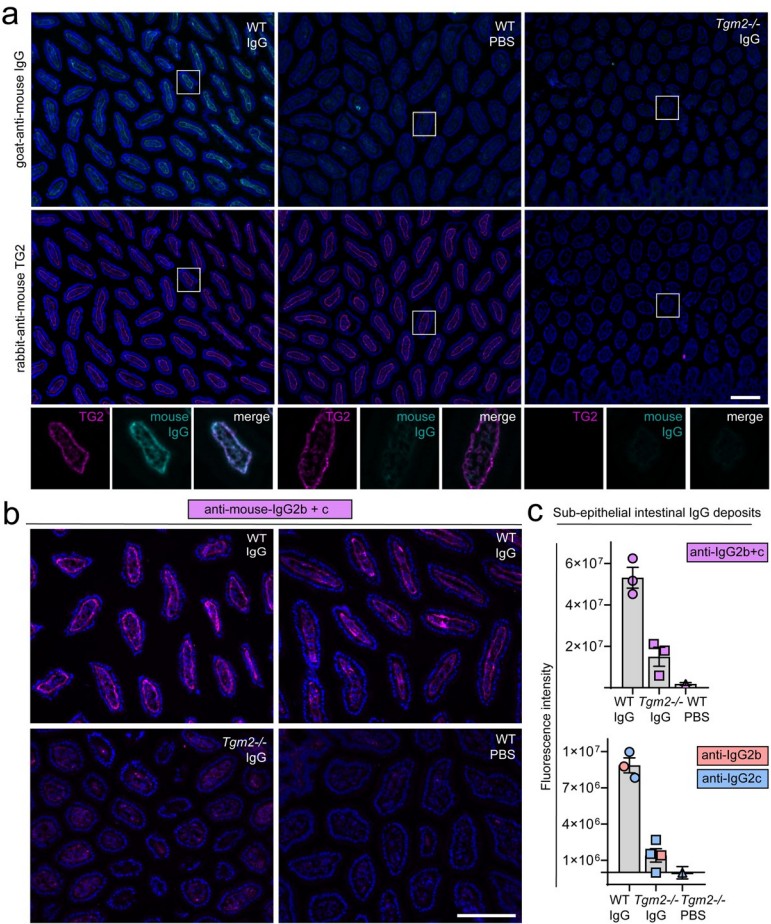

**Fig 3. Injected anti-TG2 antibodies reach the intestinal tissue.** (a) Immunofluorescence staining of frozen sections of small intestine obtained on day 20 show deposition of mouse IgG (cyan) in the basement membrane of IgG injected WT mice that co-localizes with ECM staining for endogenous TG2 (magenta). No IgG deposits were detected in PBS injected WT mice and no clear ECM deposits were observed in IgG-injected *Tgm2-/-* mice. Nuclei counterstained with DAPI are shown in blue. (b) Distribution of IgG2b and IgG2c in the small intestine of WT mice injected with IgG (top panels). Weak antibody signal is detected also in the intestine of *Tgm2*−/− mice while no signal is seen in PBS-injected mice, which indicates that antibody presence in tissue does not per se depend on presence of cognate antigen. (c) Quantification of subepithelial fluoresence signal intensity from staining for IgG2b and IgG2c together (top) or separately (bottom). Each dot represents mean fluorescence intensity calculated from one image as described in materials and methods. Bar graphs show the group mean fluorescence intensity with standard error of mean. Scale bars represent 100 μm.

intestine from WT mice injected with IgG revealed supepithelial IgG deposits that co-localized with endogenous extracellular matrix (ECM)-bound TG2 (Fig 3A). Immunofluoresence staining for mouse IgG2b and IgG2c confirmed that injected IgG reached the intestinal tissue and that ECM deposits were formed in the tissue in an antigen-dependent manner (Fig 3B and 3C). Weak signal was also observed in the small intestine of IgG-injected *Tgm2*−/− mice but no subepithelial deposits were detected. No signal was detected in PBS-injected mice.

## Clinical parameters

No signs of disease or distress were observed through the study period. The weigth gain in the different experimental groups are depicted in Figs 4 and S2. Testing weight change on day 20 of the IgG-injected WT group to each control group revealed no statistically significant

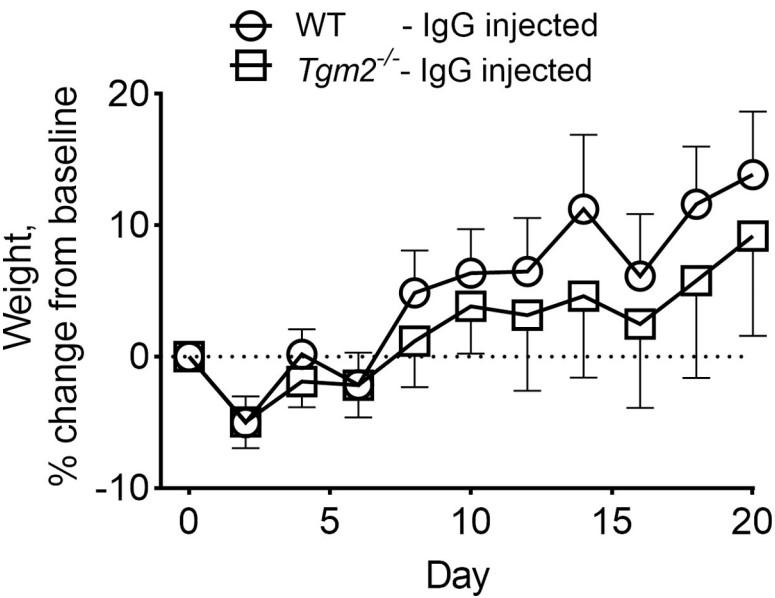

**Fig 4. Injection of anti-TG2 antibodies does not impair weight gain.** The graphs report weight as % change from baseline for WT mice and *Tgm2* deficient mice receiving injections of IgG. Dots and bars represent mean +/- SD. Data represent all mice from the two independent experiments.

differences. No obvious diarrhea occurred in any cage, although this was not evaluated in a systematic fashion.

## Tissue architechture and IEL counts

Samples of small intestine were obtained on day 20. Vh, Cd and Vh/Cd ratio were measured as demonstrated in Fig 5A. In duodenum, there was no statistically significant difference between the IgG-injected WT group and any control group (Fig 5B). In ileum, Vh and Vh/Cd ratio were slightly lower in IgG-injected *Tgm2*-/- mice compared to IgG-injected WT mice (p = 0.022 and 0.046, respectively (Fig 5C). These differences were not considered biologically relevant. Next, IELs were counted (Fig 6A). In duodenum, there were no statistically significant differences between the WT-IgG group and any of the control groups (Fig 6B). In ileum, the IEL count was significantly higher in the IgG-injected *Tgm2*-/- group compared to the IgG-injected WT group (p = 0.006, Fig 6B). However, the difference was not considered biologically relevant. Taken together, the histologic evaluation revealed no signs enteropathy in any group.

## Discussion

In this study we found no evidence for a direct role of secreted anti-TG2 in the pathogenesis of celiac enteropathy as evaluated by standard histologic criteria and clinical parameters. Our approach has several advantages compared to previous *in vivo* studies. Injecting murine immunoglobulins permits immunocompetent recipients and eliminates the need to introduce foreign proteins, virus vectors or adjuvant. Also, even though human IgGs bind mouse Fc-receptors with affinities comparable to mouse IgGs [27], the clinical outcome may still differ when species-incompatible istotypes are used.

Effector functions of IgG antibodies are mediated through their interaction with Fcγ-receptors on the cell surface and by interactions with the complement system. C57Bl/6 mice express

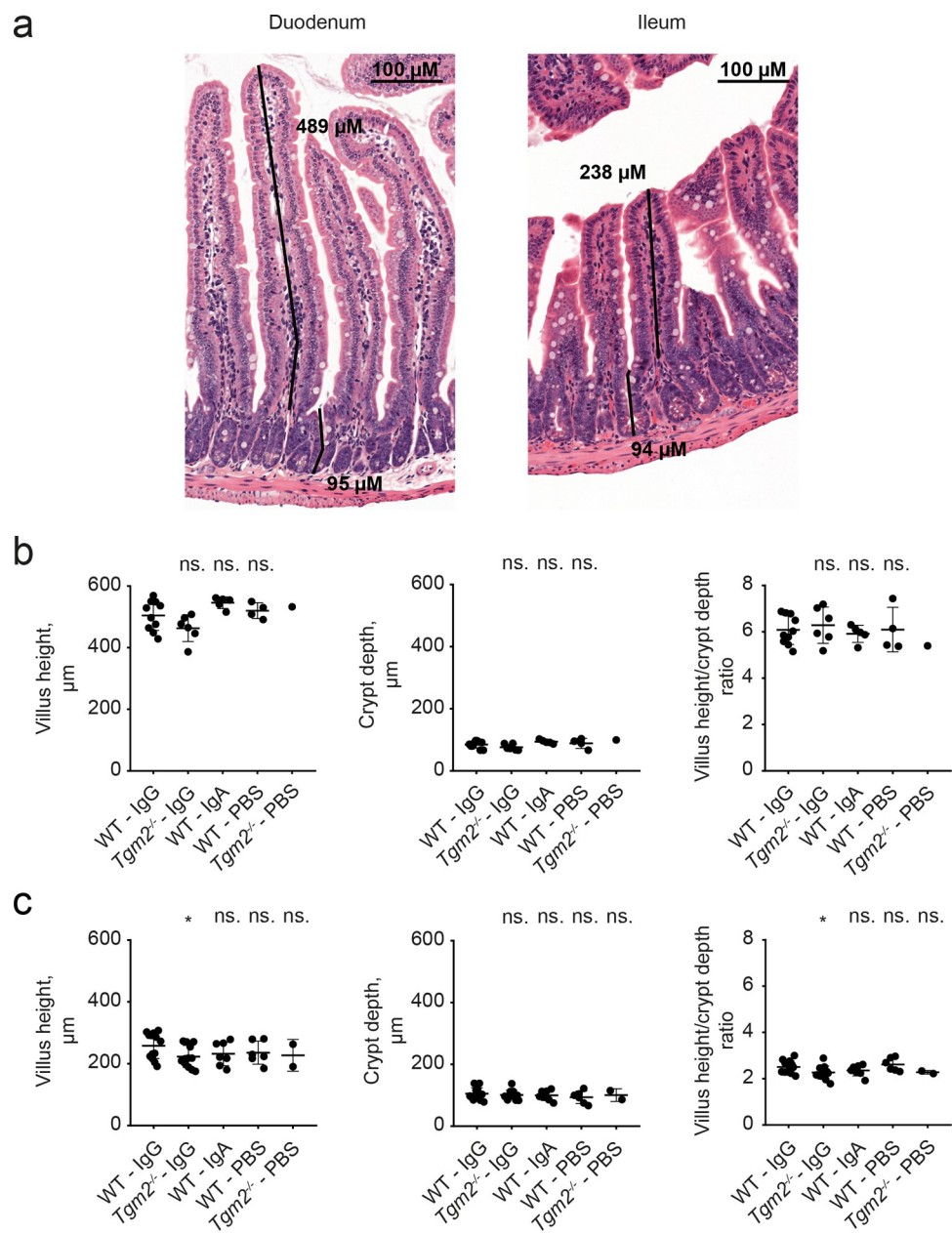

**Fig 5. No difference in mucosal architecture between groups.** (a) Examples of Vh and Cd measurements. Formalin-fixed paraffin-embedded samples stained with hematoxylin/eosin. Representative images show well-oriented pairs of villus and crypt from duodenum and ileum. Numbers indicate length of corresponding bars. (b, c) Vh, Cd and Vh/Cd ratio in duodenum (b) and ileum (c) of the different experimental groups. The WT IgG group was compared to each control group by individual Mann-Whitney tests. Data are pooled from two independent experiments. Bars represent mean +/- SD. *$P \leq 0.05$. n.s.: Not significant.

IgG1, IgG2b, IgG2c and IgG3. Of note, these are not direct homologues to the IgG subclasses in humans. IgG2 subtypes are generally considered the most potent mediators of cellular cyto-toxicity and complement activation. Although not formally characterized, IgG2c is believed to have comparable properties to IgG2a. As opposed to IgG1 and IgG3, IgG2a (and hence, proba-bly IgG2c) and IgG2b bind to all stimulatory mouse FcγRs [27,28]. Moreover, mouse IgG1 does not activate complement, and has even been implicated with anti-inflammatory

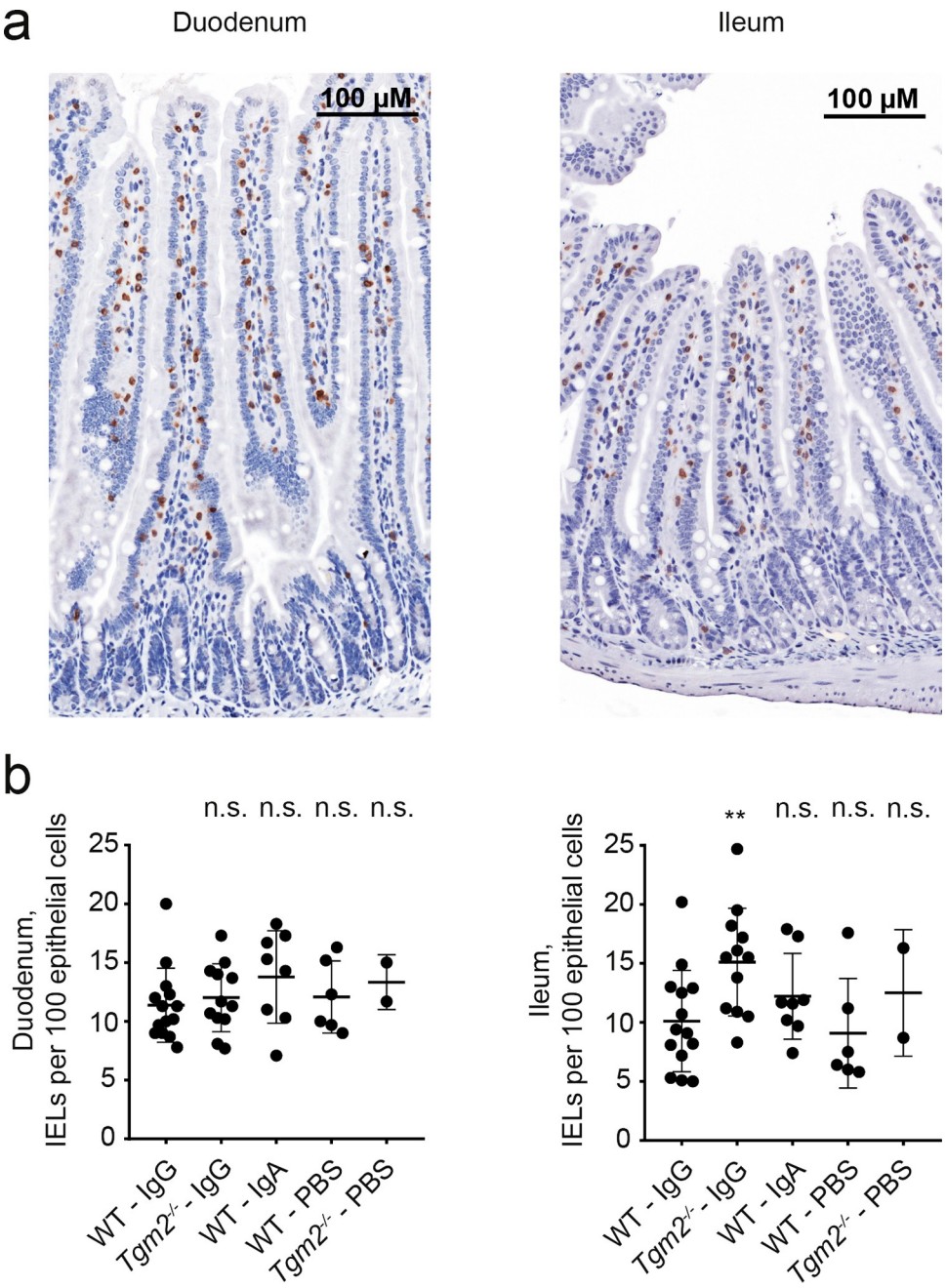

**Fig 6. IEL count in duodenum and ileum.** (a) Examples of staining for anti-CD3. Formalin-fixed paraffin-embedded sections stained with anti-CD3 and counterstained with hematoxylin. Representative images show CD3+ cells in duodenum and ileum. (b) CD3+ IELs per 100 epithelial cells in well-oriented villi in duodenum and ileum. The WT IgG group was compared to each control group by individual Mann-Whitney tests. Data represent all mice from the two independent experiments. Bars represent mean +/- SD. **P ≤ 0.01. n.s.: Not significant.

properties [29]. As such, the apporach of Di Niro et al. [14] using IgG1-based mini-antibodies may have been suboptimal.

If anti-TG2 antibodies were pathogenic, the duration of exposure necessary to develop enteropathy would be unknown. Minor effects were reported already on day eight in the

studies by Kalliokoski et al. [15,16]. Enteropathy is usually detectable in celiac patients after two weeks of gluten challenge. At this time, serum anti-TG2 is still normal or mildly increased [30–32]. However, high local concentrations in the intestinal tissue could be present earlier. We believe our trial length of 20 days would be sufficient to detect a significant contribution by anti-TG2 to enteropathy.

Immunoglobulin serves biological roles as secreted and water-soluble antibodies operating in extracellular fluids, but also as the antigen receptor of B-cells being anchored in the cell membrane as a transmembrane protein. This study is only addressing the role of anti-TG2 immunoglobulins as secreted antibodies. An involment of anti-TG2 immunoglobulins in the pathogenesis of celiac disease as B-cell receptor is likely [33]. Further, our study is also only addressing the role of anti-TG2 immunoglobulins in relation to enteropathy. Anti-TG2 immunoglobulin may have effects elsewhere in the body, effects which could very well explain many of the extraintestinal manifestations of celiac disease [34].

In a mouse model of celiac disease, B cells were found to be important in the pathogenesis by testing mice that were made devoid of B cells by genetic manipulation[35]. Yet in the same mouse model with B cells present, circulating anti-TG2 antibodies could not be detected [36]. These observations support the notion that circulating anti-TG2 antibodies are not implicated in generation of the celiac enteropathy.

Recent observations from the above mentioned animal model [36] and from a clinical trial with a TG2 inhibitor [37] support the notion that TG2 is engaged in the pathogenesis of celiac disease, and that the catalytic activity of the enzyme is involved. The antibodies that celiac disease patients make against TG2, as is the case for the 14E06 antibody, do not interfere with the catalytic activity of TG2 [17]. Observing effects of celiac patient antibodies that would implicate inhibition of enzyme activity would thus be unexpected.

Based on accumulated *in vivo* data, we believe that anti-TG2 immunoglobulins in the form of secreted antibodies do not play a major role in the development of enteropathy in celiac disease. Therefore, efforts to discover novel therapeutics are probably better directed elsewhere. Of note, anti-TG2 immunoglobulins may still play an important role in pathogenesis of celiac disease as the antigen receptor of B cells which present antigen to T cells [33]. Also, the contribution of anti-TG2 to extraintestinal manifestations of celiac disease has not been investigated in detail. This would be an interersting topic for future research.

## Supporting information

**S1 Fig. Confirmation of mouse TG2 reactivity of mAb 14E06.** Hybridoma-derived 14E06 (mouse IgG2c) binds to endogenous TG2 in the ECM of mouse small intestine (left panel). Mouse 14E06 (IgG2c) also binds to recombinant human or mouse TG2 immobilized in the ECM of *Tgm2*-/- mouse small intestine (middle and right panel). Nuclei were counterstained with DAPI. Scale bar represents 100µm.
(TIF)

**S2 Fig. Weight change in control groups.** The graph reports weight as % change from baseline for the different groups as indicated. Dots and bars represent mean +/- SD. Data represent all mice of each group from the two independent experiments.
(TIF)

**S1 Dataset.**
(XLSX)

## Acknowledgments

We thank Liv Kleppa, Bjørg Simonsen, Marie Kongshaug Johannesen and Alisa Dewan for excellent technical assistance and support. We thank the staff at the Department of Comparative Medicine, Oslo University Hospital, Rikshospitalet, for animal husbandry and care, and staff at the Department of Pathology, Oslo University Hospital (Rikshospitalet and Radium-hospitalet), for preparation of histologic samples.

## Author Contributions

**Conceptualization:** Ludvig M. Sollid.

**Formal analysis:** Christian B. Lindstad.

**Funding acquisition:** Ludvig M. Sollid.

**Investigation:** Christian B. Lindstad, Jorunn Stamnaes.

**Methodology:** M. Fleur du Pré, Ludvig M. Sollid.

**Project administration:** M. Fleur du Pré, Ludvig M. Sollid.

**Supervision:** M. Fleur du Pré, Jorunn Stamnaes, Ludvig M. Sollid.

**Validation:** Christian B. Lindstad, Jorunn Stamnaes.

**Visualization:** Christian B. Lindstad, Jorunn Stamnaes.

**Writing – original draft:** Christian B. Lindstad.

**Writing – review & editing:** M. Fleur du Pré, Jorunn Stamnaes, Ludvig M. Sollid.

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
