## [Decision Letter · Decision Letter 0]

24 Jan 2022

PONE-D-21-39201Injection of prototypic celiac anti-transglutaminase 2 antibodies in mice does not cause enteropathyPLOS ONE

Dear Dr. Lindstad,

Thank you for submitting your manuscript to PLOS ONE. After careful consideration, we feel that it has merit but does not fully meet PLOS ONE’s publication criteria as it currently stands. Therefore, we invite you to submit an extensively revised version of the manuscript that addresses all the points raised during the review process including the inclusion of additional experimental data (see issue #3 below)..

In particular three major issues have been raised by one of the reviewers.

1- "There is no discussion on the cross-reactivity between human and mouse TG2. While figure 3a shows much stronger binding of the IgG antibodies to WT than to TG2-/- intestine, IgG binding in figure 3b seems very weak. How reproducible is the binding of anti-TG2 IgG to mouse TG2 and here to mouse intestinal tissues? Can it be quantified? What is the affinity of the anti-human TG2 antibody for mouse TG2? Why is there some weak IgG binding in TG2-/- mice (as indicated by the authors). Where is it localized? Does it colocalize with TG2? Is the localization of TG2 shown in figure 3b expected? The authors have previous reported that TG2 is largely present in epithelial cells. Here binding seems to be in the basement membrane. Can these discrepancies be discussed."

2- "Since TG2 is activated by tissue damage or inflammation, is it conceivable that the effect of the ant-TG2 antibody may be different in these pathological conditions (notably in the case of IgG, if complement becomes available locally). If so, should this be specifically addressed?"

3- "Only one anti-TG2 specificity was tested. The authors indicate that they use a prototypic anti-TG2 antibody. Yet they have produced a spectrum of antibodies and described their binding to different epitopes. It is conceivable that different antibodies may have different impact. The authors have shown that, in CeD most antibodies bind to the N-terminus and interfere minimally with TG2 activity. Yet is it sufficient to test only one specificity to draw such definitive conclusions. This question is also raised by the lack of indication on the reactivity of anti-human TG2 antibodies with mouse TG2. Of note it has been suggested that anti-TG2 antibodies may participate in extra-intestinal manifestations of CeD, notably in dermatitis herpetiformis as the skin lesions contain deposits of anti-TG antibodies but no T cells. The article showing a possible role of anti-TG2 antibody in dermatitis herpetiformis used human skin grafts in immunodeficient mice (Zone et al 2011 DOI 10.4049/jimmunol.1003273)."

Please note that the revised version of the manuscript will be sent back to the original reviewers for perusal.

We look forward to receiving your revised manuscript.

Yours Sincerely,

Lucienne Chatenoud

Academic Editor

PLOS ONE

Journal Requirements:

Reviewers' comments:

Reviewer's Responses to Questions

**Comments to the Author**

1. Is the manuscript technically sound, and do the data support the conclusions?

Reviewer #1: Yes

Reviewer #2: Partly

2. Has the statistical analysis been performed appropriately and rigorously? 

Reviewer #1: Yes

Reviewer #2: N/A

3. Have the authors made all data underlying the findings in their manuscript fully available?

Reviewer #1: No

Reviewer #2: Yes

4. Is the manuscript presented in an intelligible fashion and written in standard English?

Reviewer #1: Yes

Reviewer #2: Yes

5. Review Comments to the Author

Reviewer #1: The authors present very intersting and original data concerning the potential role of anti-transglutaminase 2 secreted antibodies (anti-TG2) in vivo, in an experimental murine model. Their findings are not in favour of a major role of anti-TG2 immunoglobulins in the form of secreted antibodies in the development of enteropathy in celiac disease. This manuscript is written in an intelligible fashion and in standard english. Please note two very little form orthography remarks : lignes 129 "follwed" and 278 "disdease".

Most important, I couldn't visualise Figure 4 and Figure S1, this is why I recommend Minor Revision.

Reviewer #2: In this article, Borgen et al further explore the role of the B cell response to TG2, the celiac disease autoantigen. This enzyme plays a key role in CD by deamidating gluten peptides, a post-translational modification indispensable for the activation of DQ2/8 restricted gluten specific CD4+ T cells that are instrumental in disease pathogenesis. Strikingly, active CeD is associated with a massive intestinal plasma cell response to TG2. The latter response has been extensively characterized by the authors. They have notably provided compelling evidence that the anti-TG2 antibody response can be assimilated to a hapten-carrier B cell response where TG2 becomes recognized when bound to gluten. Abadie et al have also recently shown in a CeD mouse model that B cell depletion prevents intestinal damage. Yet it is still unclear whether anti-TG2 antibodies have a pathogenic role in CeD and if so how. On the one hand, the authors have provided strong evidence that anti-TG2 B cells may participate in CeD pathogenesis by binding gluten peptides and thereby promoting their presentation to CD4+ T cells (after endocytosis and loading into HLA-DQ molecules). On the other hand, a limited number of ancient articles suggested that anti-TG2 antibodies may exert deleterious effects directly by interacting with TG2 in the intestinal tissues, a finding that contrasts with the fact that anti-TG2 antibodies can be detected in the intestine of patients with latent CeD who have no lesions.

Here the authors investigate whether anti-TG2 antibodies can induce intestinal damage in B6 mice following intravenous injection of one the numerous anti-TG2 monoclonal antibodies that they have generated. They use the same antibody in diverse murine versions (IgG2 b , IgG2c and IgA), and they analyze the intestine after repeating the injections for 21 days.

Based on the demonstration that the injected anti-TG2 IgG antibodies can be detected in the intestine of wild type B6 but only at a much lesser degree in TG2-/- mice and that the antibodies do not induce any obvious histological intestinal lesions in WT mice, the authors conclude that anti-TG2 antibody has no direct pathogenic role in intestinal lesions.

Although this reviewer could be easily convinced that anti-TG2 antibodies do not induce directly tissue damage in the intestine, additional experiments and comments seem useful to support such a very strong and definitive conclusion

1- There is no discussion on the cross-reactivity between human and mouse TG2. While figure 3a shows much stronger binding of the IgG antibodies to WT than to TG2-/- intestine, IgG binding in figure 3b seems very weak. How reproducible is the binding of anti-TG2 IgG to mouse TG2 and here to mouse intestinal tissues? Can it be quantified? What is the affinity of the anti-human TG2 antibody for mouse TG2? Why is there some weak IgG binding in TG2-/- mice (as indicated by the authors). Where is it localized? Does it colocalize with TG2? Is the localization of TG2 shown in figure 3b expected? The authors have previous reported that TG2 is largely present in epithelial cells. Here binding seems to be in the basement membrane. Can these discrepancies be discussed.

2- Since TG2 is activated by tissue damage or inflammation, is it conceivable that the effect of the ant-TG2 antibody may be different in these pathological conditions (notably in the case of IgG, if complement becomes available locally). If so, should this be specifically addressed?

3- Only one anti-TG2 specificity was tested. The authors indicate that they use a prototypic anti-TG2 antibody. Yet they have produced a spectrum of antibodies and described their binding to different epitopes. It is conceivable that different antibodies may have different impact. The authors have shown that, in CeD most antibodies bind to the N-terminus and interfere minimally with TG2 activity. Yet is it sufficient to test only one specificity to draw such definitive conclusions. This question is also raised by the lack of indication on the reactivity of anti-human TG2 antibodies with mouse TG2. Of note it has been suggested that anti-TG2 antibodies may participate in extra-intestinal manifestations of CeD, notably in dermatitis herpetiformis as the skin lesions contain deposits of anti-TG antibodies but no T cells. The article showing a possible role of anti-TG2 antibody in dermatitis herpetiformis used human skin grafts in immunodeficient mice (Zone et al 2011 DOI 10.4049/jimmunol.1003273).

4- Are IgG relevant to the induction of lesions in the intestine of celiac patients

As indicated by the authors, conclusion concerning the IgA version of the antibody is difficult as very little IgA is detectable in the serum. Based on the fact that IgA deficient patients can develop (are prone to) celiac disease, the authors conclude that it is not important to address the role of IgA. I nevertheless wonder whether this suggestion is fully valid as it is likely that patients with IgA deficiency have IgM against TG2 in their intestine, the role of which is not addressed here. Is it clear that CeD patients have prominent anti-TG2-IgG in the intestine? Since the authors only work during 3 weeks, would it be pertinent to use hybridomas as backpack to provide continuously IgA antibodies to the mice and circumvent their short half-life.

5- Recent work shows that inhibiting TG2 has positive effects both in an animal model quoted by the authors and in humans in a recent trial (Schuppan et al 2021 10.1056/NEJMoa2032441). Is this observation pertinent to discuss the lack of deleterious role of anti-TG2 antibodies?

6- The structure of the report seems strange to this reviewer as figure legends are directly inserted in the results section and are used directly to describe the results. Is it an acceptable format in Plosone? If not, the narrative in the result section needs to be thoroughly revised and implemented.

7- The authors use parametric tests (t-tests) for statistics. Although it is a minor point as comparison between treated animals and controls does not show any difference, it is somewhat surprising to this reviewer who would have chosen non parametric tests to compare normal versus disease conditions in the absence of demonstration that data follow a normal distribution.

6. PLOS authors have the option to publish the peer review history of their article (what does this mean?). If published, this will include your full peer review and any attached files.

Reviewer #1: No

Reviewer #2: **Yes: **Nadine Cerf-Bensussan

---

## [Author Response · Author response to Decision Letter 0]

28 Feb 2022

PONE-D-21-39201 Response to reviewer comments

Response to Editors comments

The three points raised by the Editor are addressed in the response to Reviewer #2, please see below. 

Response to the reviewers’ comments

Reviewer #1:

The authors present very intersting and original data concerning the potential role of anti-transglutaminase 2 secreted antibodies (anti-TG2) in vivo, in an experimental murine model. Their findings are not in favour of a major role of anti-TG2 immunoglobulins in the form of secreted antibodies in the development of enteropathy in celiac disease. This manuscript is written in an intelligible fashion and in standard english. Please note two very little form orthography remarks : lignes 129 "follwed" and 278 "disdease". Most important, I couldn't visualise Figure 4 and Figure S1, this is why I recommend Minor Revision.

Author response:

We thank reviewer for pointing of these two typos. They are now corrected. We have ensured that Figure 4 and Figure S1 (now S2) can be fully visualized in the revised proof version.

Reviewer #2: 

In this article, Borgen et al further explore the role of the B cell response to TG2, the celiac disease autoantigen. This enzyme plays a key role in CD by deamidating gluten peptides, a post-translational modification indispensable for the activation of DQ2/8 restricted gluten specific CD4+ T cells that are instrumental in disease pathogenesis. Strikingly, active CeD is associated with a massive intestinal plasma cell response to TG2. The latter response has been extensively characterized by the authors. They have notably provided compelling evidence that the anti-TG2 antibody response can be assimilated to a hapten-carrier B cell response where TG2 becomes recognized when bound to gluten. Abadie et al have also recently shown in a CeD mouse model that B cell depletion prevents intestinal damage. Yet it is still unclear whether anti-TG2 antibodies have a pathogenic role in CeD and if so how. On the one hand, the authors have provided strong evidence that anti-TG2 B cells may participate in CeD pathogenesis by binding gluten peptides and thereby promoting their presentation to CD4+ T cells (after endocytosis and loading into HLA-DQ molecules). On the other hand, a limited number of ancient articles suggested that anti-TG2 antibodies may exert deleterious effects directly by interacting with TG2 in the intestinal tissues, a finding that contrasts with the fact that anti-TG2 antibodies can be detected in the intestine of patients with latent CeD who have no lesions.

Here the authors investigate whether anti-TG2 antibodies can induce intestinal damage in B6 mice following intravenous injection of one the numerous anti-TG2 monoclonal antibodies that they have generated. They use the same antibody in diverse murine versions (IgG2 b , IgG2c and IgA), and they analyze the intestine after repeating the injections for 21 days.

Based on the demonstration that the injected anti-TG2 IgG antibodies can be detected in the intestine of wild type B6 but only at a much lesser degree in TG2-/- mice and that the antibodies do not induce any obvious histological intestinal lesions in WT mice, the authors conclude that anti-TG2 antibody has no direct pathogenic role in intestinal lesions.

Although this reviewer could be easily convinced that anti-TG2 antibodies do not induce directly tissue damage in the intestine, additional experiments and comments seem useful to support such a very strong and definitive conclusion

1- There is no discussion on the cross-reactivity between human and mouse TG2. While figure 3a shows much stronger binding of the IgG antibodies to WT than to TG2-/- intestine, IgG binding in figure 3b seems very weak. How reproducible is the binding of anti-TG2 IgG to mouse TG2 and here to mouse intestinal tissues? Can it be quantified? What is the affinity of the anti-human TG2 antibody for mouse TG2? Why is there some weak IgG binding in TG2-/- mice (as indicated by the authors). Where is it localized? Does it colocalize with TG2? Is the localization of TG2 shown in figure 3b expected? The authors have previous reported that TG2 is largely present in epithelial cells. Here binding seems to be in the basement membrane. Can these discrepancies be discussed.

Author response: 

We thank the reviewer for bringing up these important points. We have previously demonstrated that the anti-TG2 antibody 679-14-E06 binds with high and equal affinity to human and mouse TG2 (KD about 5nM as determined by surface plasmon resonance; STable1 in du Pre et al. J Exp Med 2020; PMID31727780). This fact is now stated in the text (line 71 in the revised manuscript with track changes). We have also shown that 14-E06 expressed as human IgG1 recognizes endogenous mouse TG2 in unfixed frozen tissue-sections of intestine (Cardoso et al, FEBS 2015, PMID: 25808416). To demonstrate that hybridoma-derived 14-E06 as used in this study recognizes mouse TG2, we have included a supplementary figure that shows binding of 14-E06 mouse IgG2c to extracellular matrix bound TG2 in mouse small intestine (new S1 Figure). Thus, we are certain that the antibodies we have injected can bind to mouse TG2 with high affinity. 

IgG deposits were reproducibly observed in the sub-epithelial extracellular matrix of wild-type (WT) mice injected with IgG. This signal co-localized with endogenous extracellular matrix bound TG2, as shown in our revised Figure 3a. IgG deposits were seen in all tissue sections analyzed from WT mice injected with IgG from which we had fresh frozen intestine (n = 4) (revised Figure 3b and c). Some variation in signal intensity within tissue sections likely reflect regional differences in IgG tissue concentration, whereas variation in overall signal staining intensity depends on the secondary antibody used for staining (revised Figure 3c). Presence of some IgG signal in Tgm2-/- mice injected with IgG suggests that antibody presence in the tissue is independent of cognate antigen presence. However, no extracellular IgG deposition was observed in absence of TG2. This has been clarified in line 196-197 and in the legend of Figure 3. From quantification of the sub-epithelial fluorescence signal (as per reviewer’s suggestions) we see a marked difference between IgG-injected WT and Tgm2-/- mice (revised Materials and Methods, and revised Figure 3c). As alluded to by the reviewer, the true in vivo localization of TG2 remains unknown as there is a clear discrepancy between the lack of B cell tolerance to TG2 (du Pre et al. J Exp Med 2020; PMID31727780) and the abundant extracellular TG2 staining that is observed in frozen tissue sections. We are currently working to resolve this discrepancy, and we do not consider this to be a scope of this paper. 

2- Since TG2 is activated by tissue damage or inflammation, is it conceivable that the effect of the ant-TG2 antibody may be different in these pathological conditions (notably in the case of IgG, if complement becomes available locally). If so, should this be specifically addressed?

Author response: 

We agree that an effect of complement activation cannot be excluded, and that both complement factors, TG2 and anti-TG2 IgG may be present in highly inflamed tissue such as in untreated celiac disease (PMID: 1537512). We have in our study addressed whether presence of anti-TG2 antibodies may play a role in development of enteropathy as has previously been suggested (PMID: 25209899, PMID: 23824706, PMID: 27503559). We find no evidence to support this notion. 

3- Only one anti-TG2 specificity was tested. The authors indicate that they use a prototypic anti-TG2 antibody. Yet they have produced a spectrum of antibodies and described their binding to different epitopes. It is conceivable that different antibodies may have different impact. The authors have shown that, in CeD most antibodies bind to the N-terminus and interfere minimally with TG2 activity. Yet is it sufficient to test only one specificity to draw such definitive conclusions. This question is also raised by the lack of indication on the reactivity of anti-human TG2 antibodies with mouse TG2. Of note it has been suggested that anti-TG2 antibodies may participate in extra-intestinal manifestations of CeD, notably in dermatitis herpetiformis as the skin lesions contain deposits of anti-TG antibodies but no T cells. The article showing a possible role of anti-TG2 antibody in dermatitis herpetiformis used human skin grafts in immunodeficient mice (Zone et al 2011 DOI 10.4049/jimmunol.1003273).

Author response: 

We have conclusively demonstrated that the antibody (14-E06) we have injected strongly binds to mouse TG2 (see response to #1). While the anti-TG2 response in celiac disease indeed can target multiple epitopes on TG2, 14-E06 represents the group of antibodies most frequently shared across patients (ref 17, PMID: 22366952). If anti-TG2 antibodies should play a role in development of intestinal remodeling, such an effect must be mediated by public antibodies shared across patients. Therefore, we consider mAb 14-E06 to be well suited to address this question. We do however agree with the reviewer that antibodies targeting other epitopes may have different effects on TG2. As we also state in our manuscript (line 300), we do not exclude a role for TG2 antibodies in the development of extra intestinal manifestations, but this warrants further investigation. 

4- Are IgG relevant to the induction of lesions in the intestine of celiac patients

As indicated by the authors, conclusion concerning the IgA version of the antibody is difficult as very little IgA is detectable in the serum. Based on the fact that IgA deficient patients can develop (are prone to) celiac disease, the authors conclude that it is not important to address the role of IgA. I nevertheless wonder whether this suggestion is fully valid as it is likely that patients with IgA deficiency have IgM against TG2 in their intestine, the role of which is not addressed here. Is it clear that CeD patients have prominent anti-TG2-IgG in the intestine? Since the authors only work during 3 weeks, would it be pertinent to use hybridomas as backpack to provide continuously IgA antibodies to the mice and circumvent their short half-life.

Author response: 

As long as we see IgG making tissue deposits in frozen sections of mice intestine, the specific antibody is indeed present and should be able to exert effects. The suggestion to introduce IgA hybridomas in vivo to secure continuous supply is interesting but we foresee issues to control equal production in different animals. Further we are unsure whether this approach would indeed allow us to prolong the time of exposure since visible tumors are present already 9-13 days after injection (PMID 9042427) and humane endpoints (i.e. maximum allowable tumor size) are expected to be reached shortly thereafter. In addition to these concerns, the timeline to obtain ethical approvals and finalize such studies would be long making it impossible to complete these studies within the scope of this paper.

5- Recent work shows that inhibiting TG2 has positive effects both in an animal model quoted by the authors and in humans in a recent trial (Schuppan et al 2021 10.1056/NEJMoa2032441). Is this observation pertinent to discuss the lack of deleterious role of anti-TG2 antibodies?

Author response: 

We thank the reviewer for this suggestion. We agree with the reviewer that the recent observations in an animal model of celiac disease and a clinical trial of man strongly support the notion that TG2 is implicated in the pathogenesis of celiac disease. We have introduced a small paragraph on this point in the Discussion (line 309).

6- The structure of the report seems strange to this reviewer as figure legends are directly inserted in the results section and are used directly to describe the results. Is it an acceptable format in Plosone? If not, the narrative in the result section needs to be thoroughly revised and implemented.

Author response:

We confirm that this indeed is the desired format of PLoS ONE. 

7- The authors use parametric tests (t-tests) for statistics. Although it is a minor point as comparison between treated animals and controls does not show any difference, it is somewhat surprising to this reviewer who would have chosen non parametric tests to compare normal versus disease conditions in the absence of demonstration that data follow a normal distribution.

Author response:

We agree that a non-parametric test could be appropriate in this setting, and we thank the reviewer for pointing this out. In the revised version of the manuscript, the non-parametric Mann-Whitney test has been used for all the comparisons reported. Importantly, our conclusions remain unchanged with this test.

---

## [Editor Report · Decision Letter 1]

23 Mar 2022

Injection of prototypic celiac anti-transglutaminase 2 antibodies in mice does not cause enteropathy

PONE-D-21-39201R1

Dear Dr. Lindstad,

We’re pleased to inform you that the revised version of your manuscript has been judged scientifically suitable for publication and will be formally accepted once it meets all outstanding technical requirements.

Yours Sincerely,

Lucienne Chatenoud

Academic Editor

PLOS ONE
---

## [Editor Report · Acceptance letter]

28 Mar 2022

PONE-D-21-39201R1 

Injection of prototypic celiac anti-transglutaminase 2 antibodies in mice does not cause enteropathy

Dear Dr. Lindstad:

I'm pleased to inform you that your manuscript has been deemed suitable for publication in PLOS ONE. Congratulations! Your manuscript is now with our production department. 

Kind regards, 

on behalf of

Professor Lucienne Chatenoud 

Academic Editor

PLOS ONE